# Proposed Method for the Design of Geosynthetic-Reinforced Pile-Supported (GRPS) Embankments

**Rashad Alsirawan \*, Edina Koch and Ammar Alnmr**

Department of Structural and Geotechnical Engineering, Széchenyi István University, 9026 Győr, Hungary
\* Correspondence: alsirawan.rashad@sze.hu

**Abstract:** Soft soils with unfavorable properties can be improved using various ground-improvement methods. Among these methods, geosynthetic-reinforced pile-supported (GRPS) embankments are considered a reliable option for challenging ground conditions and time-bound projects. Nevertheless, the intricate load transfer mechanism of the GRPS embankment presents challenges due to the multiple interactions among its components. To overcome the limitations of current design methods that do not fully account for all interactions, a simplified design method has been developed for GRPS embankments. This method uses numerical analysis to predict pile load efficiency and geosynthetic tension. In this study, a validated model of the GRPS embankment, which incorporates certain simplifications for design purposes, was adopted. Based on this simplified model, a database of load efficiency and geosynthetic tension was collected to derive the design equations. The design method employed six parameters, namely, pile cap width, pile spacing, embankment height, oedometric modulus of the subsoil, geosynthetic stiffness, and embankment fill unit weight. The design process utilized Plaxis 3D and Curve Expert software. The results showed reasonable agreement between the findings of the proposed design method and the field measurements of eight case studies.

**Keywords:** simplified design method; load efficiency; geosynthetic tension; geosynthetic-reinforced pile-supported embankments

## 1. Introduction

Geosynthetic-reinforced pile-supported (GRPS) embankments are increasingly being used to provide support for railways, roads, and highways built on weak soils [1]. The use of geosynthetic reinforcement has been shown to increase loads transfer towards piles, resulting in benefits such as the reduction of required piles and the expedition of the construction process. Furthermore, incorporating geosynthetic layers underneath conventional pile-supported (CPS) embankments can address significant settlements, a low bearing capacity of the subsoil, and lateral displacement at the embankment toe [2,3].

By using fewer piles and reducing the use of concrete, GRPS embankments can help reduce embodied carbon (EC), which refers to the carbon dioxide released during the production, transportation, construction, and disposal of materials. As a result, GRPS embankments offer a sustainable and cost-effective solution for infrastructure development on weak soils [3].

This integrated system transfers loads resulting from the embankment weight and surcharge to a firm bearing soil layer. The load transmitted in a GRPS embankment is partitioned into three segments: (i) direct transmission to the piles via the soil arch, (ii) transmission to the piles by means of geosynthetic reinforcement, and (iii) transfer to the soft subsoil. The arching phenomenon observed in the first load segment can be attributed to the relatively higher stiffness of the piles compared to the subsoil. The second load segment induces two phenomena: the geosynthetic experiences membrane tension, and the geosynthetic–soil interface experiences frictional interaction. The third load segment results in the subsoil exerting an upward counter pressure known as "subsoil support" [1,4].

Several design methods are proposed to study the behavior of GRPS embankments. These methods belong to different generations. Terzaghi [5] developed the arching theory based on the trapdoor test. He also observed that shear forces are developed along the frictional surfaces due to differences in stiffness. Russell and Pierpoint's design method [6] used Terzaghi's concept to describe the behavior of the supported embankments, depending on a three-dimensional numerical model. In this method, the support from the soft subsoil is disregarded, and the entire embankment load is borne by the piles. Guido et al. [7] conducted several plate loading tests. This design method assumes that under three-dimensional conditions, the load distributes in the embankment body, forming a pyramid with an angle of $45°$, and the geosynthetic carries the pyramid weight completely. Hewlett and Randolph [8] performed a set of three-dimensional model tests in which the soil arch was described as a hemi-spherical dome. This design method assumes that the limit states can occur in the arch crown or in the pile caps. Based on the assumptions of this method, the subsoil support is not considered. Zhang et al. [9] extended the soil-arching concept, representing the hemi-spherical domes as covering the triangular configurations of the piles. Hewlett and Randolph's design method was developed by Low et al. [10] based on a set of laboratory tests. This design method asserts that the arch shape is semi-cylindrical over walls of piles, and the soft subsoil carries part of the applied loads. Russel et al. [11] suggested a new design method using a finite difference analysis of a three-dimensional model in which the subsoil support is considered. Three-dimensional model tests were also carried out by Kempfert et al. [12] to develop a design method which assumes that a multi-shell dome represents the soil arch. The subsoil support is taken into account in this design method. The design method by Collin et al. [13] improved the approach of Guido et al. by creating a stiffened load transfer platform LTP (acting as a beam) with numerous geosynthetic layers to redirect a significant part of the load into the piles. The Nordic Geosynthetic Group [14] theorized that the shape of the soil supported by geosynthetic layers acts as a soil wedge in the design method. A theoretical design method using a semi-circular soil arch was proposed by Abusharar et al. [15]. This method includes the effect of the uniform surcharge and subsoil support, but it is considered mathematically complex.

In the BS 8006 design method [16], two formulas were used to compute the load imparted to the geosynthetic. One was the formula of Marston, which was developed based on plane strain tests on flexible culverts for high embankments; the other formula was Hewlett and Randolph's formula, which assumes that the soil arch is a hemi-spherical dome. Due to groundwater fluctuations, the subsoil support does not count in this method. Van Eekelen et al. [17] changed the amount of load borne by the geosynthetic in the case of Marston's formula. The EBGEO design method [18], which employs a triangular load distribution on the geosynthetic, utilizes the principles of Kempfert et al. [14]. An analytical model was proposed by Van Eekelen et al. [19] for designing GRPS embankments. The arch model is more complicated than other methods and can be described as a model of concentric hemispheres. This method is thought to be conceptually and mathematically challenging for use in design. Both EBGEO [18] and Van Eekelen et al. [19] take the support of subsoil into account but with a linear model.

Zhuang et al. [20] introduced a model to analyze the piled embankment, building upon the work of Hewlett and Randolph. This method proposed some refinements of the failure mechanisms in the crown and pile caps. The design method outlined in the Dutch standard CUR 226 [21] incorporates the arch model developed by Van Eekelen et al. [19]. This method assumes an inverted, triangular distribution of load on the geosynthetic layer when the subsoil provides no support, while a uniform distribution of load is assumed when subsoil support exists. Filz et al. [22] proposed a method of load displacement compatibility (LDC) to analyze GRPS embankments. This method proposed some refinements to Terzaghi's method to estimate the soil arch. The subsoil support is considered in this method. Pham's design method [1] combined the arching theory and the membrane theory. This method proposes the effect of the subsoil in two models (a linear model and a non-linear model). In

addition, the frictional interactions between the geosynthetic layers and the surrounding soil are considered in the analysis.

Based on a review of the literature, none of the previously mentioned methods are able to precisely describe the behavior of a GRPS embankment. The moot points are noticeable, such as the soil arch model, load distribution over the geosynthetic layer(s), the role of the soft subsoil, and the subsoil behavior, if it is considered. This is due to the proposed simplifications in each design method, which are attributed to plenty of interactions between the elements of this system.

The aim of this paper is to propose a more comprehensive approach to design, using the finite element method (FEM). The characteristic of this method is the ability to consider the real behavior of underlying soft soil, the frictional behavior of the geosynthetic–soil interface, the consolidation of the soft soil, and all the expected interactions for which the other methods have not been able to complete this mission to different degrees. A validated model of a GRPS embankment with some simplifications is used to perform this study, which focuses on deriving two equations for determining the load efficiency of the piles and the tension in the geosynthetic. Additionally, the proposed design method is evaluated by comparing field measurements from a series of GRPS embankments with the results obtained from these equations to assess their validity.

## 2. Proposed Design Method

The aim of the method presented in this study is to propose a new concept of design that can avoid some of the drawbacks of experimental and theoretical methods. These analytical design methods differ in the determination of the shape of the soil arch, the magnitude of load applied to the geosynthetic reinforcement, the skin friction at the soil–geosynthetic interface, and the role of the soft subsoil and its real behavior. These variations in the analysis result in a significant difference between the outcomes of these analytical methods. Therefore, the FE method is widely regarded as an effective approach to addressing these challenges and achieving a dependable design.

The incorporation of numerical modeling in the design process contributes to ending the controversy over the moot points regarding the reciprocal behavior between the elements of this system. However, similar to former design methods and in order to avoid complicating the issue, this method is not able to include all the parameters related to embankment fill, soft soil, piles, and geosynthetic reinforcement because of their large number. For the purpose of the design, the following key parameters should be determined: (i) the load efficiency of the pile ($E$), to derive the pile bearing capacity; (ii) the tension in the geosynthetic ($T$), to derive the strain and vice versa.

Drawing on the aforementioned approach, the proposed design method employs numerical analysis as its fundamental basis. The following steps outline the methodology employed in the presented design approach:

1.  A full-scale model of the GRPS embankment, which included a wide range of field measurements, is described in this study as the starting point.
2.  To ensure the reliability of the numerical model, a calibration and validation process was conducted with a high level of accuracy.
3.  For design purposes, simplifications were made to the numerical model of the GRPS embankment. These simplifications were carefully selected to reduce computational complexity and improve the efficiency of the analysis while still providing comprehensive and dependable results that are suitable for design.
4.  Based on the simplified numerical model of the GRPS embankment, a wide database of the parameters ($E$; $T$) was collected through an extended parametric study. The database was analyzed using a mathematical technique to predict two equations that can be used to calculate the key parameters of E and T. Figure 1 shows the key steps of the proposed design method.

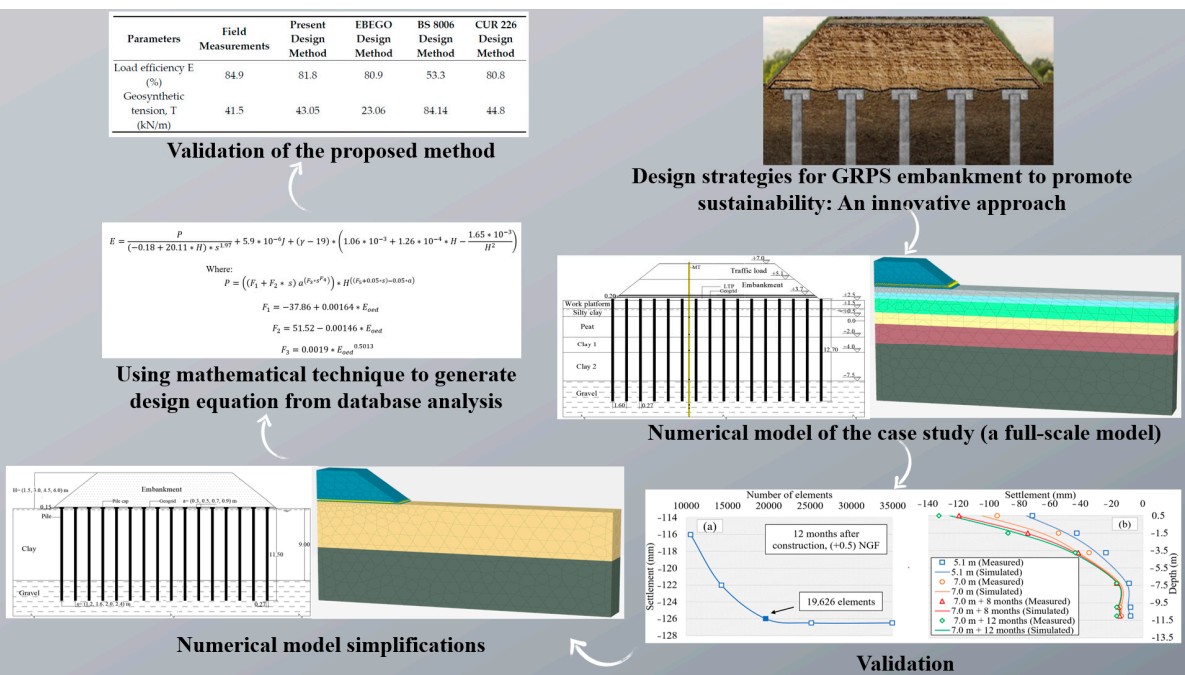

**Figure 1.** Methodology for a proposed design method.

## 2.1. Description of GRPS Embankment Model

A full-scale GRPS embankment was built over multiple layers of soft soil (silty clay, peat, and two layers of clay). The soil characterization was completed with six cone penetration tests, three boreholes, and a comprehensive suite of laboratory tests. The groundwater table was located 1.1 m under the surface of working platform, which was built prior to the construction stage to ease the equipment movement. A network of pre-cast concrete piles was driven inside the soft soil to reach a firm layer of gravel located beneath the multiple soft soil layers. The cross-sectional area of the pile was equal to 0.0751 m$^2$, and the pile spacing was equal to 1.6 m. The LTP was constructed to be 0.7 m thick, and two uniaxial geogrid layers were inserted inside the LTP and located (0.2; 0.4) m over the pile heads. To simulate both the embankment body and the traffic load, the embankment was built in two stages, each with a thickness of 1.9 m. The embankment was located at Virvée swamp as a part of a high-speed railway line that connects Tours and Bordeaux in France [23,24]. Figure 2 depicts the cross-section of the GRPS embankment model. The soil levels were linked to a predetermined zero point (French georeferenced level (*NGF*)) [23].

## 2.2. Numerical Modeling of GRPS Embankment: Calibration and Validation

The simulation of numerous complicated issues now necessitates the use of numerical modeling. The FE method is a powerful numerical modeling tool that is frequently used in geotechnical engineering [25,26]. Within this study, a new design method for a GRPS embankment is provided based on FE analysis, relying on the merits of "the finite element program".

In this article, the Plaxis 3D program was used for the validation process of the GRPS embankment. To simulate the behavior of the embankment fill, soft soil layers, and gravel layer, the hardening soil (*HS*) model was utilized. Additionally, the piles were represented as embedded beam elements, while the geogrid was modeled using elastoplastic material [24]. To reduce the impact of the boundaries on the accuracy of the simulation, the authors extended the side boundary of the model by 45 m along the x-axis. The model had a depth of 18 m from the ground surface and was fixed in both the vertical and horizontal directions at the bottom, which prevented excessive displacement during testing. Additionally, the vertical boundaries were constrained horizontally to prevent any potential distortion of the results. A mesh sensitivity analysis was applied to choose the

mesh size. A medium mesh distribution was used in the simulations, as the settlement behavior was approximately the same in fine and medium distributions. Based on this choice, the number of mesh triangular elements, with 10 nodes and an average size of 0.828 m, was 19,626, as depicted in Figure 3a. This type of mesh element is commonly used in Plaxis 3D simulations to provide a precise representation of the soil and structure geometry. The triangular shape allows for the modeling of complex geometries, and the 10 nodes ensure a high level of precision in the computation of the soil–structure interaction.

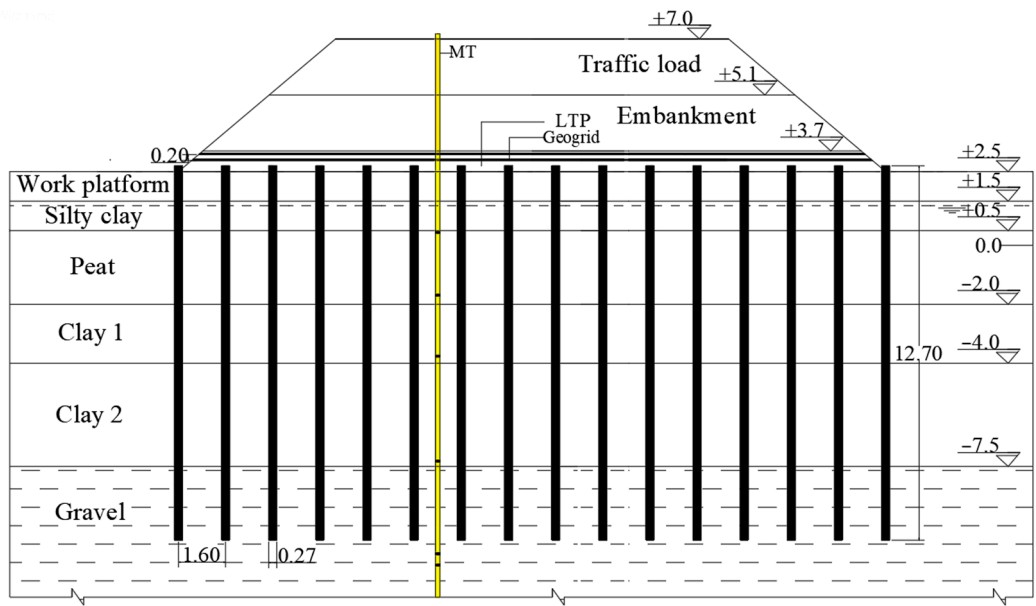

**Figure 2.** Cross section of GRPS embankment (dimensions in meters)—representative drawing [23].

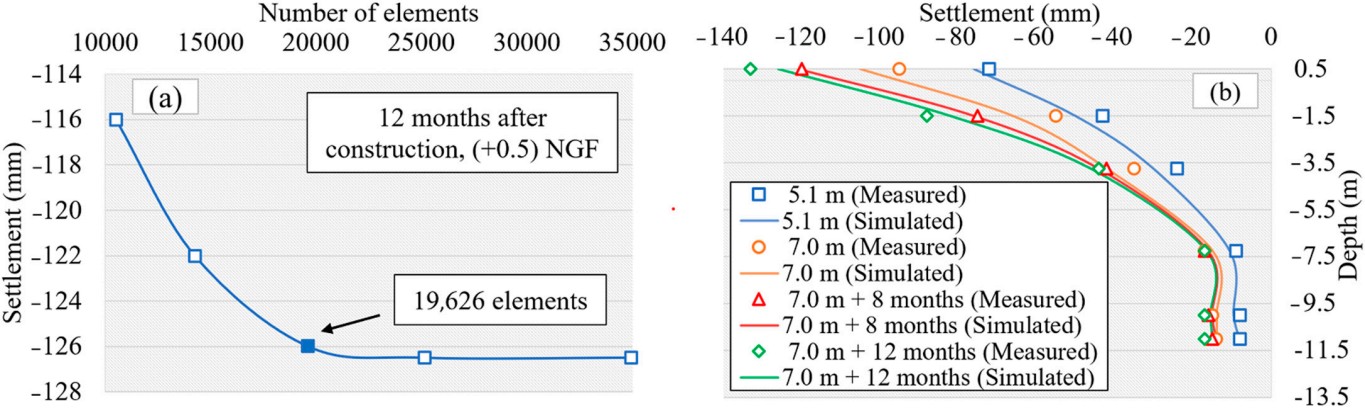

**Figure 3.** (**a**) The mesh sensitivity analysis. (**b**) Comparison between predicted and measured settlements.

The calibration process involved identifying the optimal values of model parameters to minimize the difference between the numerical predictions and actual measurements. To achieve this, a back analysis approach was employed to optimize the parameters. The initial set of parameters resulted in predicted settlements that were lower than those observed in the field. Through the back analysis, the stiffness parameters of the hardening soil model were refined to produce a better match between the predicted and measured values.

For the validation process, a comparison was made between the settlements predicted from the simulation and those measured in the field with the use of six magnet rings that were fixed on a magnetic probe extensometer (*MT*). The settlements were recorded underneath the embankment body at different depths of 0.5, −1.5, −3.75, −7.25, −10, and −11 m (*NGF*). The field measurements were performed throughout the following stages (Figure 3b): (i) the end of the embankment stage, (ii) the end of the traffic load stage, (iii) at

8 months, and (iv) 12 months after the construction of the full-scale model. Furthermore, another comparison was performed between the vertical stresses measured using the earth pressure cells ($EPC_i$) and the results from the FE analysis at the pile head ($EPC_1$) and over the load transfer platform ($EPC_2$) (see Figure 4). The soil-arching phenomenon resulted in a concentration of stress at the pile head. The instruments in the LTP and the soft soil are illustrated in Figure 2. To obtain additional information regarding the validation process of the full-scale model and the properties of the soft soil layers, embankment fill, geogrid layer, and piles, refer to reference [24].

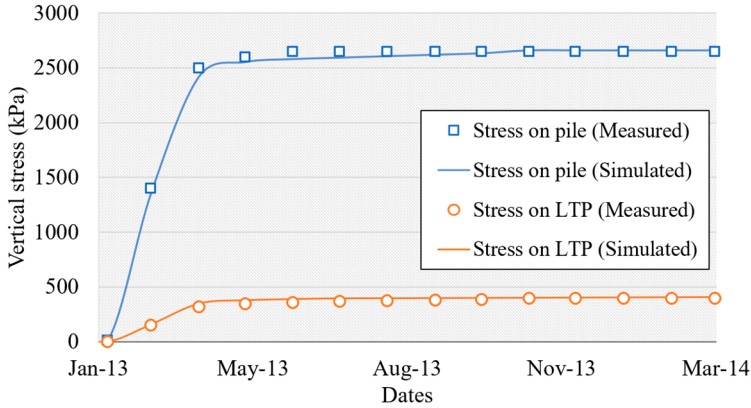

**Figure 4.** Comparison between predicted and measured vertical stresses.

### 2.3. A Simplified Model of the Validated GRPS Embankment

Once the validation process for the essential full-scale model was complete, simplifications were made to the validated GRPS embankment model in order to facilitate its generalization. The model was subsequently simplified as follows:

- One normal, consolidated clayey soil layer with a thickness of 9 m was used instead of multiple over-consolidated soft soil layers, as in the original model. The properties of the gravel, clay, and embankment soils are summarized in Table 1.
- The working platform was removed. This is consistent with the majority of the case studies, as noted in the literature.

**Table 1.** Summary of soil properties.

| Basic Parameters | Characters and Units | Embankment Fill | Clay | Gravel |
|---|---|---|---|---|
| Material model | - | Hardening soil | Hardening soil | Hardening soil |
| Unsaturated unit weight | $\gamma_{unsat}$ (kN/m$^3$) | 19 | 13.5 | 19 |
| Saturated unit weight | $\gamma_{sat}$ (kN/m$^3$) | 21 | 14.5 | 20 |
| Internal friction angle | $\varphi^{\circ}$ | 35 | 29 | 35 |
| Dilatancy angle | $\Psi^{\circ}$ | 5 | 0 | 5 |
| Cohesion | $c$ (kPa) | 5 | 4 | 10 |
| Reference secant stiffness | $E_{50}^{ref}$ (kN/m$^2$) | 30,000 | 800 | 63,000 |
| Reference tangent stiffness | $E_{oed}^{ref}$ (kN/m$^2$) | 30,000 | 450 | 63,000 |
| Reference unloading–reloading stiffness | $E_{ur}^{ref}$ (kN/m$^2$) | 90,000 | 2400 | 189,000 |
| Exponential power | $m$ | 0.5 | 0.6 | 0.5 |
| Coefficient of earth pressure at rest | $K_0^{nc}$ | 0.426 | 0.515 | 0.426 |
| Unloading/reloading Poisson's ratio | $v_{ur}$ | 0.2 | 0.2 | 0.2 |
| Failure ratio | $R_f$ | 0.9 | 0.9 | 0.9 |
| Permeability | $k$ (m/day) | 0.864 | $5.55 \times 10^{-4}$ | 1.00 |
| Over consolidation ratio | $OCR$ | - | 1 | - |

Figure 5a illustrates a representative drawing of the GRPS embankment model subsequent to undergoing simplifications, whereas Figure 5b displays the FE mesh of the same model.

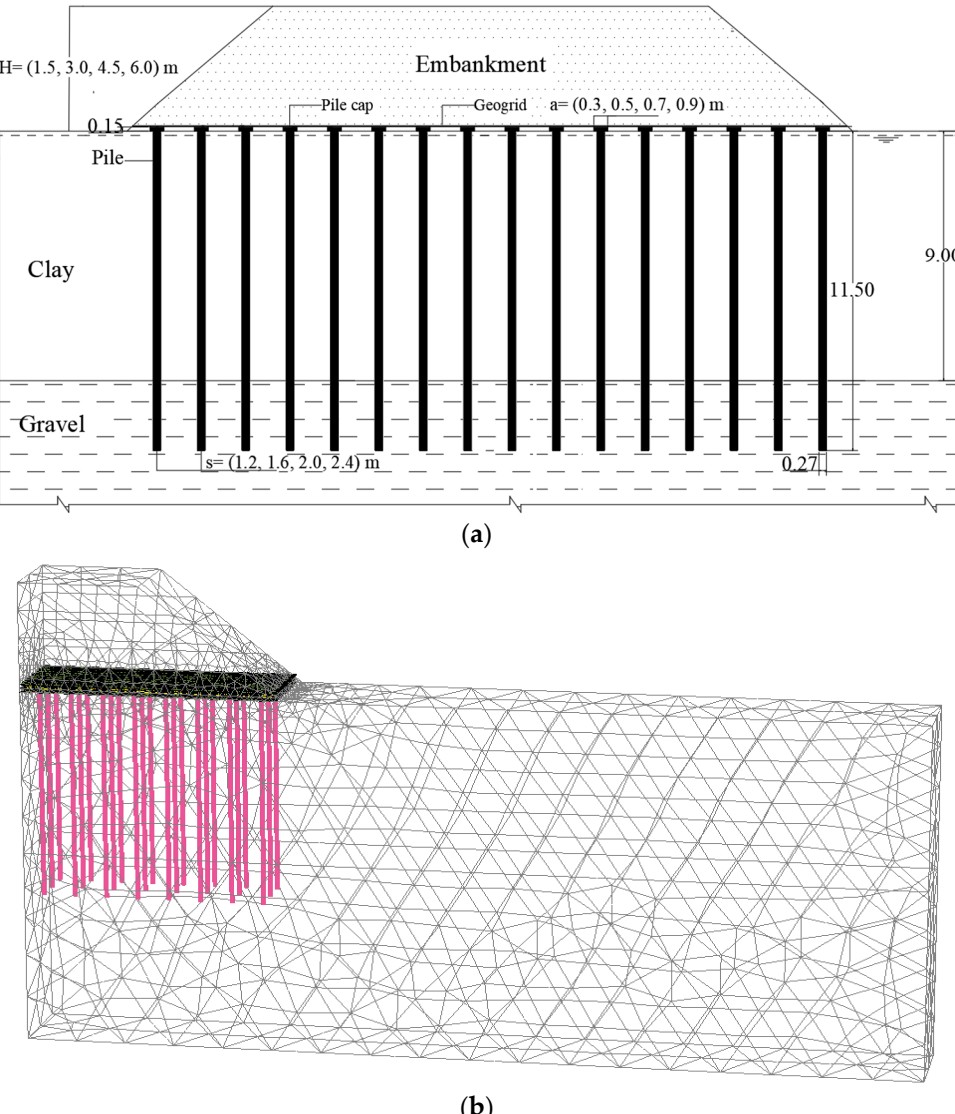

**Figure 5.** (**a**) Cross section of the simplified GRPS embankment (dimensions in meters); (**b**) FE mesh of the simplified GRPS embankment.

The cover ratio can be governed by adjusting the pile volume or by enlarging the pile cap. The load transfer mechanism is different in two cases. In the first case, the increase in the pile volume simultaneously contributes to an increase in skin friction and a decrease in load efficiency. In the second case, the enlargement of the pile cap leads to an increase in soil arching and consequently an increase in load efficiency, as indicated in Figure 6, while the pile volume is fixed. The tension in the geosynthetic is affected to a small extent due to the change in the load transfer mechanism, as illustrated in Figure 7. The present design method assumes that the pile diameter (a) is fixed at 0.3 m, and the cover ratio can be controlled by the pile cap surface area.

One layer of the biaxial geogrid was utilized in the simplified model. It was observed that inserting one layer of geogrid within the LTP enhances the load efficiency by about 60%, while inserting two layers of geogrid increases the load efficiency by about 63% when compared to an unreinforced piled embankment [27]. In order to achieve a higher load efficiency, the layer of geogrid was positioned directly over the pile heads, which is considered a cost-effective and efficient solution [27]. Table 2 presents the characteristics of the pile material and geogrid.

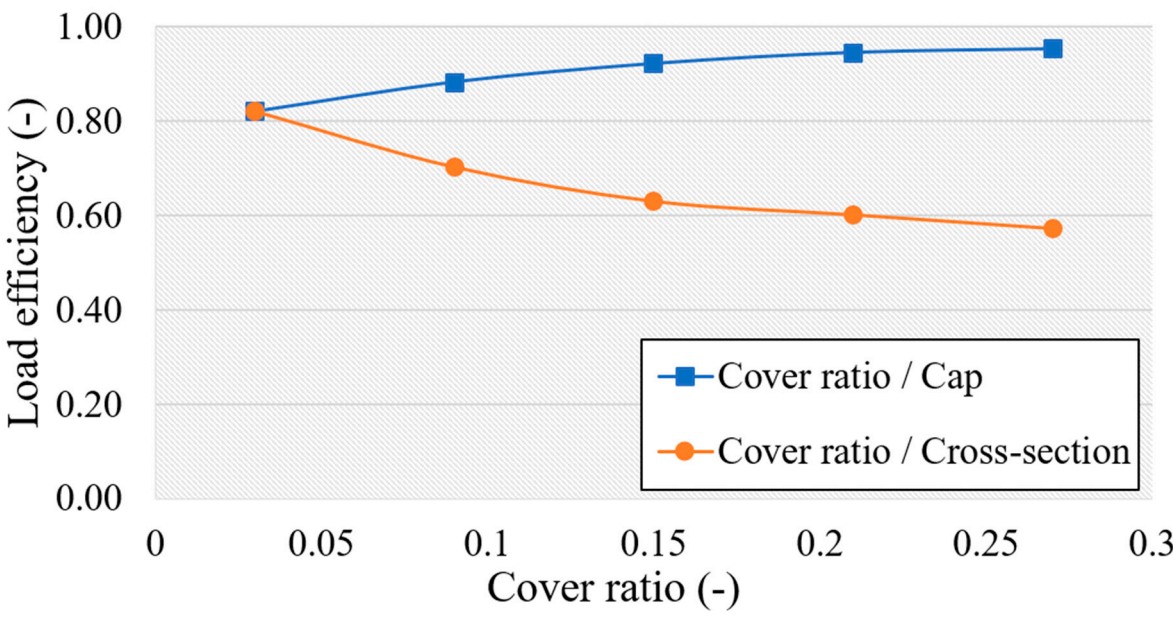

**Figure 6.** Influence of cover ratio on load efficiency in the two cases.

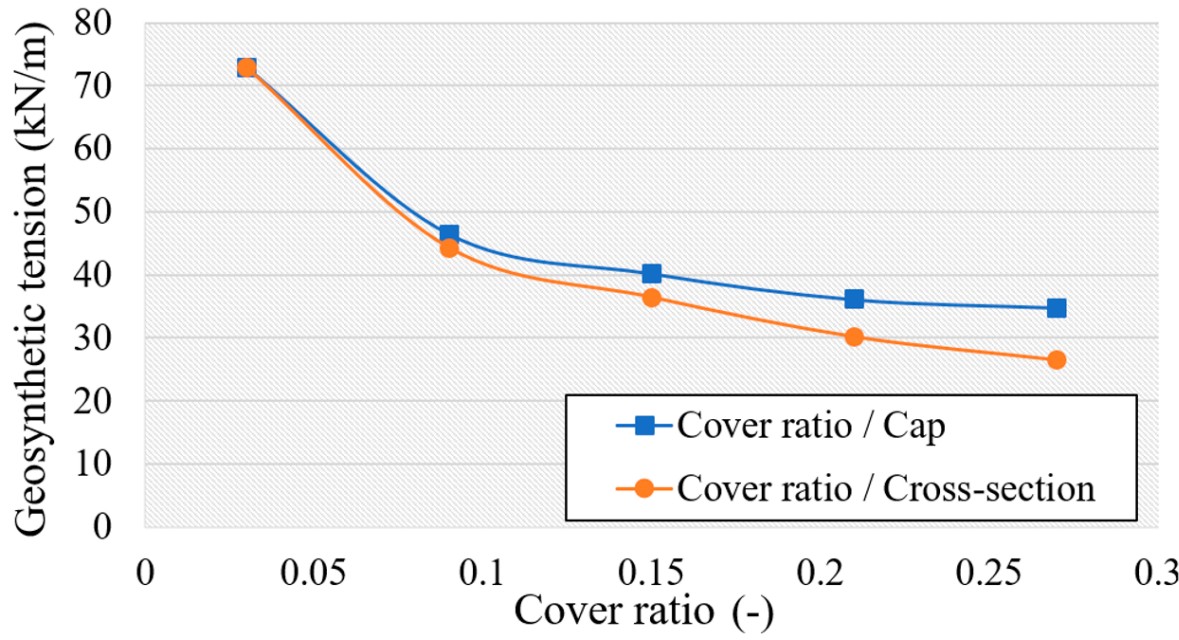

**Figure 7.** Influence of cover ratio on geosynthetic tension in the two cases.

**Table 2.** Summary of the pile and geogrid properties.

|  | Unit Weight $\gamma$ (kN/m$^3$) | Young Modulus $E$ (GPa) | Poisson Ratio $v$ (-) | Stiffness $J$ (kN/m) | Length $L$ (m) |
|---|---|---|---|---|---|
| Pile | 24 | 20 | 0.2 | - | 11.5 |
| Geogrid | - | - | 0.2 | 13,000 | - |

*2.4. Mathematical Derivation of the Design Equations*

The present method has been developed based on an extended parametric study obtained from the simplified numerical model. In the beginning of the process, data were collected, including the load efficiency and geosynthetic tension. We ran the Plaxis 3D program over 1000 times with four embankment heights in every calculation process, which was adequate to provide a precise prediction of the values of the dependent variables (*E*;

*T*). The second step was the analysis of the data in order to derive equations that could be used to predict the values of the dependent variables. These equations were based on a set of the following independent variables: pile cap width (*a*), pile spacing (*s*), embankment height (*H*), oedometric modulus of subsoil ($E_{oed}$), geosynthetic stiffness (*J*), and unit weight of the embankment fill ($\gamma$). By using the Curve Expert program, a regression model was constructed to describe the interdependence of the independent and dependent variables. Based on the probability concept, the final number of observations for each dependent and independent variable was $4^6 = 4096$. Table 3 contains the presumed values of the independent variables.

**Table 3.** Presumed values of the independent variables.

| | |
|---|---|
| Pile cap width (*a*), m | 0.3, 0.5, 0.7, 0.9 |
| Pile spacing (*s*), m | 1.2, 1.6, 2.0, 2.4 |
| Embankment height (*H*), m | 1.5, 3.0, 4.5, 6.0 |
| Subsoil stiffness ($E_{oed}$) kN/m$^2$ | 1000, 4000, 7000, 10,000 |
| Geosynthetic stiffness (*J*) kN/m | 1000, 5000, 9000, 13,000 |
| Unit weight of embankment fill ($\gamma$) kN/m$^3$ | 17.0, 19.0, 21.0, 23.0 |

In this methodology, a data structure tree was used in which the root represented the subsequent Equations (9) and (10) of the dependent variables (*E*; *T*). There were five levels of parent nodes representing independent variables (*H*; $E_{oed}$; *J*; *a*; *s*). The last level of the parent nodes ended in leaf nodes, which represented the last independent variable ($\gamma$). The nodes at every level are called siblings [28]. The siblings are integrated in every level to produce a new generation of parents. Herein, in the work mechanism to derive the equation of geosynthetic tension (*T*) with the following constants (*a*; $E_{oed}$; *J*; *s*), the unit weight of the embankment fill ($\gamma$) affects the geosynthetic tension (*T*), according to the following equation:

$$T = A + B * \Delta\gamma \tag{1}$$

where *A*; *B* represent the coefficients in Equation (1), and $\Delta\gamma = (\gamma - 19)$.

For different values of the embankment height (*H*), the geosynthetic tension (*T*) is changed in Equation (1) as shown in Table 4.

**Table 4.** Influence of embankment height (*H*) on the geosynthetic tension (*T*).

| | |
|---|---|
| *H* = 1.5 m | $T_1 = A_1 + B_1 * (\gamma - 19)$ |
| *H* = 3.0 m | $T_2 = A_2 + B_2 * (\gamma - 19)$ |
| *H* = 4.5 m | $T_3 = A_3 + B_3 * (\gamma - 19)$ |
| *H* = 6.0 m | $T_4 = A_4 + B_4 * (\gamma - 19)$ |

To include the impact of the height of embankment (*H*) in the equation of the geosynthetic tension (*T*), the relationships between *A* and *B* (as dependent variables) and *H* (as independent variable) were found using the Curve Expert program, which are provided as follows:

$$A = F_1 * F_2{}^H \tag{2}$$

$$B = G_1 + G_2 * H + G_3 * H^2 \tag{3}$$

where $F_i$; $G_i$ represent the coefficients in Equations (2) and (3).

Substituting Equations (2) and (3) into Equation (1), an expression of the geosynthetic tension (*T*) in terms of the height of embankment (*H*) and embankment fill unit weight ($\gamma$) is provided:

$$T = F_1 * F_2{}^H + \left(G_1 + G_2 * H + G_3 * H^2\right) * (\gamma - 19) \tag{4}$$

To determine how the pile cap width (a) affects the geosynthetic tension (T), the relationships between $F_i$ and $G_i$ (as dependent variables) and a (as independent variable) were found using the Curve Expert program, and the following expression is provided:

$$T = \left(I_1 * exp^{\frac{I_2}{a}}\right) * F_2{}^H + (G_1 + G_2 * H + G_3 * H^2) * (\gamma - 19) \tag{5}$$

Similarity, to include the effect of the pile spacing (s) on geosynthetic tension (T), the following equation is used:

$$T = \left((X_1 * exp^{(X_2*s)}) * exp^{\left(\frac{\frac{X_3}{1+X_4*exp^{X_5*s}}}{a}\right)}\right) * (X_6 + X_7 * s)^H + (G_1 + G_2 * H + G_3 * H^2) * (\gamma - 19) \tag{6}$$

Likewise, to find the effect of the geosynthetic stiffness (J) on the geosynthetic tension (T):

$$T = \left(\left((Y_1 + Y_2 * J) * exp^{((Y_3+Y_4*J)*s)}\right) * exp^{\left(\frac{\frac{Y_5*J^{Y_6}}{1+(Y_7*J^{Y_8})*exp^{(Y_9*J^{Y_{10}})*s}}}{a}\right)}\right) * (X_6 + X_7 * s)^H + (G_1 + G_2 * H + G_3 * H^2) \\ *(\gamma - 19) \tag{7}$$

The final step is to determine the effect of the elastic modulus ($E_{oed}$):

$$T = \left(\left((Y_1 + Y_2 * J) * exp^{((Y_3+Y_4*J)*s)}\right) * exp^{\left(\frac{\frac{Y_5*J^{Y_6}}{1+(Y_7*J^{Y_8})*exp^{(Y_9*J^{Y_{10}})*s}}}{a}\right)}\right) * (X_6 + X_7 * s)^H + \\ (G_1 + G_2 * H + G_3 * H^2) * (\gamma - 19) \tag{8}$$

where:

$$D_1 = Y_1 + Y_2 * J$$

$$D_2 = ((Y_3 + Y_4 * J) * s) = ((Z_1 * E_{oed}^{Z_2} + (Z_3 + Z_4 * E_{oed}) * J) * s)$$

$$D_3 = \frac{\frac{Y_5 * J^{Y_6}}{1+(Y_7*J^{Y_8}) * exp^{(Y_9 * J^{Y_{10}}) * s}}}{a} = \frac{\frac{(Z_5+Z_6 * E_{oed}) * J^{Y_6}}{1+((Z_7+Z_8 * E_{oed}) * J^{Y_8}) * exp^{((Z_9+Z_{10} * E_{oed}) * J^{Y_{10}}) * s}}}{a}$$

$$D_4 = (X_6 + X_7 * s)^H$$

where $I_i$; $X_i$; $Y_i$; $Z_i$ represent the coefficients in Equations (5)–(8).

Section 5 provides a detailed derivation of the equation used to calculate the load efficiency of the pile (E).

## 3. Results and Discussion

Initially, this section presents the equations pertaining to load efficiency and geosynthetic tension. Subsequently, a comparison is made between the results obtained from the proposed design method and the field measurements.

### 3.1. Equations of Load Efficiency and Geosynthetic Tension

The basic equations of this study are (9) and (10), which are worded using the aforementioned methodology, provided in the following equations:

- Load efficiency (E):

$$E = \frac{P}{(-0.18 + 20.11 * H) * s^{1.97}} + 5.9 * 10^{-6}J + (\gamma - 19) * \left(1.06 * 10^{-3} + 1.26 * 10^{-4} * H - \frac{1.65 * 10^{-3}}{H^2}\right) \tag{9}$$

where:

$$P = \left((F_1 + F_2 * s) a^{(F_3 * s^{F_4})}\right) * H^{((F_5+0.05 * s)-0.05 * a)}$$

$$F_1 = -37.86 + 0.00164 * E_{oed}$$

$$F_2 = 51.52 - 0.00146 * E_{oed}$$

$$F_3 = 0.0019 * E_{oed}^{0.5013}$$

$$F_4 = 12.07 * E_{oed}^{-0.2776}$$

$$F_5 = 0.993 + 6.7 * 10^{-6} * E_{oed}$$

- Geosynthetic tension ($T$):

$$T = D_1 * exp^{D_2} * exp^{D_3} * D_4 + (\gamma - 19) * \left(0.064 * (H + 1.093)^2\right) \qquad (10)$$

where:

$$D_1 = 0.078 + 6.25 * 10^{-5} * J$$

$$D_2 = \left(4.95 * E_{oed}^{-0.18} - \left(3.95 - 2.54 * 10^{-4} * E_{oed}\right) * 10^{-5} * J\right) * s$$

$$D_3 = \left(\frac{C_1/C_2}{a}\right)$$

$$C_1 = \left(1.132 - 1.63 * 10^{-5} * E_{oed}\right) * J^{-0.14}$$

$$C_2 = 1 + \left(\left(6.167 + 4.09 * 10^{-4} * E_{oed}\right) * J^{0.25}\right) * exp^{-(3.62 - 1.93 * 10^{-5} * E_{oed}) * s * J^{0.0275}}$$

$$D_4 = (1.55 - 0.05 * s)^H$$

To obtain the results, the following input data must be provided with the respective units: pile cap width ($a$), meters; pile spacing ($s$), meters; embankment height ($H$), meters; oedometric modulus of the subsoil ($E_{oed}$), kN/m$^2$; geosynthetic stiffness ($J$) kN/m; and embankment fill unit weight ($\gamma$), kN/m$^3$.

There are many fundamental issues that are considered obvious with this design method:

- This method is regarded as eligible for low- and medium-rise embankments ($H$ = 0.5–6 m).
- The cover ratio can be changed by adjusting the pile cap area, and the ratio (a/s) should be less than 0.75.
- The values of ($E$; $T$) are calculated after complete consolidation and a degree of consolidation ($U$ = 1). However, creep behavior is neglected in the design.
- The effect of surcharge load ($q$) can be compensated for using the formula: $H = q/\gamma$. Based on this, the new height of the embankment ($H$) must be used in the calculation.
- ASIRI recommendations [29] classify the soil based on the pressiometric modulus for soft soil ($E_M \leq 3$ MPa). In the limitations of the earlier classification, $E_{50} \approx 2\ E_{oed}$ was used for soft soil [30].
- In the present design method, the minimum subsoil oedometric modulus is 0.3 MPa.
- The piles are configured in squared and rectangular patterns. In the case of a rectangular pile arrangement, the equivalent width of the squared area between the piles is proposed to calculate the load efficiency; however, the length of the rectangular area between the piles is utilized to calculate the geosynthetic tension.

Pham introduced the "active depth" concept in which the oedometric modulus of the subsoil can be calculated as follows [31]:

$$E_{oed} = k_s * D_{act} \tag{11}$$

where $k_s$ is the subgrade reaction modulus (kPa/m) and $D_{act}$ is the active depth at which the vertical stress caused by the embankment fill weight and surcharge load is less than 20% of the vertical geostatic stress. It can be calculated using the following simplified Equation [31]:

For clay:

$$D_{act} = 10 * \sqrt[4]{\frac{(s - a)}{6}} \tag{12}$$

In case of various subsoil layers, the oedometric modulus can be calculated by:

$$E_{oed} = \frac{E_{oed.1} * D_1 + E_{oed.2} * D_2 + \cdots + E_{oed.n} * D_n}{D_1 + D_2 + \cdots + D_n} \tag{13}$$

where $E_{oed.1}; E_{oed.2}; E_{oed.n}$ are the oedometric moduli of the several layers, and $D_1; D_2; D_n$ are the thicknesses of the subsoil layers [31].

### 3.1.1. Sensitivity Analysis

A sensitivity analysis was conducted to assess the impact of the input parameters in Equations (9) and (10) on the load efficiency of the piles (*E*) and the geosynthetic tension (*T*). For this purpose, each parameter was increased by 40% while the other parameters remained constant. In the analysis, the following parameters were used ($s = 2$ m; $a = 0.3$ m; $H = 4.0$ m; $E_{oed} = 5.0$ MPa; $J = 6000$ kN/m; $\gamma = 18$ kN/m$^3$). The results in Table 5 demonstrate the percentage change of (*E; T*) according to the corresponding increase (40%) of the aforementioned input parameters. It is evident that the most influential parameter on (*E; T*) is the pile spacing (*s*).

**Table 5.** The sensitivity analysis.

| Parameter | $s$ (m) | $a$ (m) | $H$ (m) | $E_{oed}$(MPa) | $J$ (kN/m) | $\gamma$ (kN/m$^3$) |
|---|---|---|---|---|---|---|
| Modified value | 2.8 | 0.42 | 5.6 | 7.0 | 8400 | 25.2 |
| Percentage change of $E$ (%) | −24.8 | 9.2 | 4.0 | −5.7 | 2.5 | 5.6 |
| Percentage change of $T$ (%) | 91.1 | −26.6 | 82.0 | −9.8 | 13.1 | 36.7 |

### 3.1.2. Comparison of the Proposed Method Solutions and FE Results

Figure 8a,b compare the outputs of Plaxis 3D (dashed lines) to the results of the design method (solid lines). A remarkable agreement is found, which demonstrates that the proposed design method can effectively predict the values of load efficiency and geosynthetic tension.

### 3.2. Validation of the Proposed Method

For validation, the field measurements of a series of eight case studies under the limitations of the design method were used. The full-scale models are described briefly in this section. To extend the comparison process, The German standard (EBGEO) [18], Dutch standard (CUR 226) [21], and British standard (BS 8006)—on the basis of Hewlett and Randolph's formula [16]—design methods were inserted in the comparison. A simplified version of these design methods can be found in the literature.

### 3.2.1. GRPS Motorway Embankment in the Netherlands (Van Eekelen et al., 2020)

As part of the junction rehabilitation between motorway A12 and a local road near Woerden, Netherlands, a new motorway exit was constructed on a reinforced piled em-

bankment, according to a study by Van Eekelen et al. [32]. Shortly after the construction, it was observed that the subsoil support had become negligible. Two geogrid layers with a stiffness (*J*) of 4611 kN/m were installed within the LTP. A summary of the site conditions and the design parameters is provided herein: embankment height (*H*) = 1.96 m; unit weight ($\gamma$) = 18.3 kN/m³; surcharge load (*q*) = 4.2 kPa; pile spacing (*s*) = 2.25 m; pile cap width (a) = 0.85 m. A 17 m thick layer of soft clay and peat made up the soft subsoil, and the oedometric modulus was $E_{oed}$ = 300 kN/m².

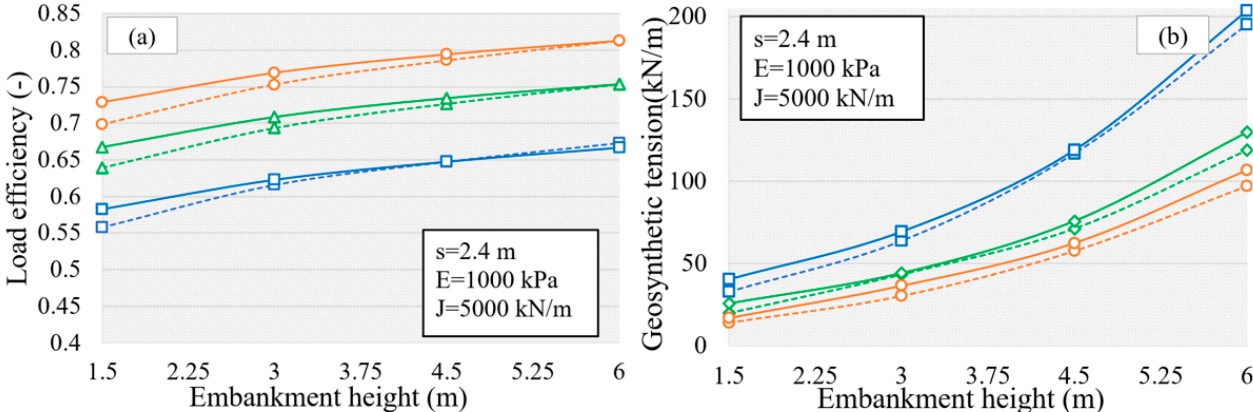

**Figure 8.** Comparison of Plaxis 3D and equation results: (**a**) load efficiency; (**b**) geosynthetic tension. Dashed line represents the outputs of Plaxis 3D; solid line represents the predicted values of the equation. For blue lines, a = 0.3 m; green lines, a = 0.5 m; orange lines, a = 0.7 m.

Table 6 shows a comparison of the present method with three design methods and the actual field measurements for load efficiency and geosynthetic tension. The results indicate a relatively close agreement with the field measurements. The EBGEO method provides a reasonably accurate prediction of load efficiency; however, it falls short in its estimation of geosynthetic tension compared to the actual measurements. On the contrary, the BS 8006 method overestimates the geosynthetic tension while providing an inaccurate prediction of load efficiency. In comparison, the CUR 226 method's predictions of both load efficiency and geosynthetic tension align well with the field measurements.

**Table 6.** Comparison of the field measurements with design method results for the Van Eekelen et al. case study.

| Parameters | Field Measurements | Present Design Method | EBEGO Design Method | BS 8006 Design Method | CUR 226 Design Method |
|---|---|---|---|---|---|
| Load efficiency *E* (%) | 84.9 | 81.8 | 80.9 | 53.3 | 80.8 |
| Geosynthetic tension, *T* (kN/m) | 41.5 | 43.05 | 23.06 | 84.14 | 44.8 |

### 3.2.2. GRPS Railway Embankment in the Netherlands (Duijnen et al., 2010)

Compressible soils are widespread in the Netherlands. The GRPS embankment was used in this project for a smooth transition between a viaduct with zero settlements and an unsupported embankment. The LTP included three geosynthetic layers with a stiffness (*J*) = 5237 kN/m. The embankment was constructed using recycled concrete material: embankment height (*H*) = 2.6 m and unit weight ($\gamma$) = 18.3 kN/m³. The subsoil consisted of two layers of sand and one of soft clay, with an oedometric modulus ($E_{oed}$) = 3300 kN/m² and a thickness of (*D*) = 8 m. The pile configuration was rectangular in shape. The spacing between the piles was 1.45 × 1.9 m², and the width of pile cap (a) = 0.4 m [31,33,34].

The load efficiency and geosynthetic tension are compared in Table 7. The present and CUR 226 methods produce results that are compatible with the measured values. The EBGEO design method relatively underestimates the load efficiency while overestimating

the geosynthetic tension relative to the field measurements. The BS 8006 design method underestimates the load efficiency while drastically overestimating geosynthetic tension.

**Table 7.** Comparison of the field measurements with design method results for the Duijnen et al. case study.

| Parameters | Field Measurements | Present Design Method | EBEGO Design Method | BS 8006 Design Method | CUR 226 Design Method |
|---|---|---|---|---|---|
| Load efficiency $E$ (%) | 76.3 | 76.5 | 70.6 | 63.7 | 76.3 |
| Geosynthetic tension, $T$ (kN/m) | 17.6 | 15.1 | 52.37 | 160.92 | 16.6 |

### 3.2.3. GRPS Highway Embankment in China (Chen et al., 2010)

A GRPS embankment in the southeastern part of Zhejiang Province in China was utilized to elevate the *TJ* (Taizhou Jinyun) highway above numerous layers of soft soil. In this project, pre-stressed tube piles with square caps and an embankment fill consisting of a mixture of clayey soil and gravel were used. A summary of the site condition and the design parameters is provided herein: stiffness of the used geogrid layer ($J$) = 1500 kN/m; embankment height ($H$) = 6 m; unit weight ($\gamma$) = 21.0 kN/m$^3$; oedometric modulus of the subsoil ($E_{oed}$) = 3322 kN/m$^2$; thickness ($D$) = 30 m; pile spacing ($s$) = 2.0 m; width of pile cap (a) = 1.0 m [35].

Table 8 presents a comparison of the measured and calculated load efficiency and geosynthetic tension. The present method, as well as BS 8006 and CUR 226 to a lesser extent, achieve results that are close to the field measurements for load efficiency. However, the tension in the geosynthetic was not recorded during the field measurements. The comparison of the various design methods reveals some variation in the results in this case study.

**Table 8.** Comparison of the field measurements with design method results for the Chen et al. case study.

| Parameters | Field Measurements | Present Design Method | EBEGO Design Method | BS 8006 Design Method | CUR 226 Design Method |
|---|---|---|---|---|---|
| Load efficiency $E$ (%) | 87.6 | 88.4 | 77.6 | 86.3 | 82.2 |
| Geosynthetic tension, $T$ (kN/m) | - | 27.75 | 36 | 47.8 | 17.6 |

### 3.2.4. Full-Scale Model of GRPS Embankment in Korea (Lee et al., 2019)

The performance of a GRPS embankment was evaluated using a full-scale model constructed in a 3 m high, 5 m wide, and 15 m long concrete test box. To simulate the soft subsoil, the voids between the model piles were filled with 0.4 m of polyurethane foam. At a height of 15 cm above the pile caps, one layer of geotextile with a stiffness ($J$) = 422 kN/m was installed. The oedometric modulus of the polyurethane foam was ($E_{oed}$) = 1510 kN/m$^2$. The properties of the poorly- to well-graded embankment fill were as follows: unit weight ($\gamma$) = 20.2 kN/m$^3$, embankment height ($H$) = 2.55 m, pile spacing ($s$) = 1.2 m, and finally, width of pile cap (a) = 0.4 m [36].

Table 9 compares the load efficiency and geosynthetic tension. According to the results, the proposed method aligns with the load efficiency measurement results. The EBGEO, CUR 226 and BS 8006 methods underestimate the load efficiency to different degrees. The comparison of the design methods for geosynthetic tension in Table 9 shows some differences in the findings. However, it should be noted that no field measurements were made for geosynthetic tension.

### 3.2.5. GRPS Highway Embankment in China (Zhao et al., 2019)

The Jin-Bin highway is located in China. A GRPS embankment was proposed to increase the width of the current highway by 5 m on both sides and to extend the four two-way lanes into six or eight two-way lanes. A rectangular arrangement of piles was

used in the construction, with a transversal pile spacing ($s_1$) = 2.5 m and a longitudinal pile spacing ($s_2$) = 4.5 m. The pile cap width (a) = 1.0 m. On top of the pile caps, a 0.4 m thick crushed stone layer was laid, into which two geogrid layers with a stiffness ($J$) = 1630 kN/m were inserted. The soil profile consisted of seven layers of soft soil in which the oedometric modulus of these layers ($E_{oed}$) = 8254 kN/m$^2$, thickness ($D$) = 13.5 m, embankment height ($H$) = 4.8 m, and unit weight ($\gamma$) = 19.0 kN/m$^3$ [37].

**Table 9.** Comparison of the field measurements with design method results for the Lee et al. case study.

| Parameters | Field Measurements | Present Design Method | EBEGO Design Method | BS 8006 Design Method | CUR 226 Design Method |
|---|---|---|---|---|---|
| Load efficiency *E* (%) | 76.4 | 79.3 | 57.4 | 67.6 | 63.6 |
| Geosynthetic tension, *T* (kN/m) | - | 4.79 | 25.32 | 46.8 | 12.74 |

A comparison between the results of load efficiency and geosynthetic tension obtained from measurements and calculations is provided in Table 10. The current method accurately reflects the measured load efficiency. On the other hand, the EBGEO, CUR 226, and BS 8006 methods show an underestimation of the load efficiency. The geosynthetic tension was not recorded in the measurements. The comparison between the analytical design methods reveals disparities in the results, however, with the BS 8006 method presenting a significantly higher tension in the geosynthetic compared to the other methods.

**Table 10.** Comparison of the field measurements with design method results for the Zhao et al. case study.

| Parameters | Field Measurements | Present Design Method | EBEGO Design Method | BS 8006 Design Method | CUR 226 Design Method |
|---|---|---|---|---|---|
| Load efficiency *E* (%) | 66.6 | 64.1 | 29.0 | 37.2 | 35.4 |
| Geosynthetic tension, *T* (kN/m) | - | 69.9 | 97.8 | 1210.5 | 15.2 |

3.2.6. GRPS Highway Embankment in China (Liu et al., 2015)

An embankment over a group of grouted gravel piles with two geogrid layers was constructed to support a highway in China. The embankment's fill material was a cohesive soil mixed with approximately 40% fly ash. The soil profile comprised four layers of a silty clay, mud clay, soft clay, and medium silty clay with a full thickness of 27.5 m. Two geogrid layers were installed over the pile caps. A summary of the input data required for the design method is provided herein. The embankment height ($H$) = 4.6 m, the unit weight ($\gamma$) = 19 kN/m$^3$. The oedometric modulus of the subsoil ($E_{oed}$) = 2196 kN/m$^2$. The pile spacing (s) = 2.4 m and the pile cap width (a) = 1.0 m. the geogrid stiffness ($J$) = 1125 kN/m [31,38].

Table 11 reveals the outcomes of comparing several design methods with the field measurements, demonstrating that the present method provides load efficiency results that are relatively consistent with the field measurements. The EBGEO method underestimates the load efficiency, while the BS 8006 and CUR 226 methods provide results that are relatively close to the measurements. In this study, the tension in the geosynthetic was not included in the measurements. When various design methods are compared, it becomes apparent that the results differ from one another; however, the present approach is in agreement with the CUR 226 method.

3.2.7. GRPS Stockyard Embankment in Brazil (Hosseinpour et al., 2015)

In Rio de Janeiro, Brazil, an embankment supported by a network of geotextile-encased granular columns and one layer of geogrid was constructed to support a stockyard, which is used to store raw materials and coal for steel plate manufacturing. The aim of this project was to decrease the settlement and horizontal displacement and reduce the construction time. Hosseinpour et al. [38] reported the details of the design parameters

and site conditions: geogrid stiffness ($J$) = 2000 kN/m, embankment height ($H$) = 5.35 m), unit weight ($\gamma$) = 27.5 kN/m$^3$, oedometric modulus of the subsoil ($E_{oed}$) = 5813 kN/m$^2$, thickness ($D$) = 11 m, pile spacing ($s$) = 2.0 m, and pile diameter without cap ($d$) = 0.7 m [39].

**Table 11.** Comparison of the field measurements with design method results for the Liu et al. case study.

| Parameters | Field Measurements | Present Design Method | EBEGO Design Method | BS 8006 Design Method | CUR 226 Design Method |
|---|---|---|---|---|---|
| Load efficiency *E* (%) | 77.7 | 82.4 | 62.8 | 71.4 | 68.8 |
| Geosynthetic tension, *T* (kN/m) | - | 20.67 | 48.4 | 105.8 | 23.6 |

Although this case study is out of the limits of the present design method due to the fact that the cover ratio is governed by the pile diameter and not by the cap width, the geosynthetic tension should be very close to the real value measured in the field while the load efficiency is radically different, as previously explained in Figures 6 and 7. Table 12 compares the load efficiency and tension in the geosynthetic. The EBGEO, CUR 226, and BS 8006 methods produce excessively conservative results in terms of load efficiency. The present method slightly overestimates the geosynthetic tension, which is consistent with the results in Figure 7 in which the geosynthetic tension is somewhat higher in the case of a cover ratio controlled by the pile diameter than in the case of a cover ratio controlled by the cap surface area. The EBGEO method is relatively in line with the measured tension in geosynthetic; however, the BS 8006 method significantly overestimates the geosynthetic tension, while the CUR 226 method underestimates it.

**Table 12.** Comparison of the field measurements with design method results for the Hosseinpour et al. case study.

| Parameters | Field Measurements | Present Design Method | EBEGO Design Method | BS 8006 Design Method | CUR 226 Design Method |
|---|---|---|---|---|---|
| Load efficiency *E* (%) | 23.4 | - | 81.8 | 82.2 | 84.8 |
| Geosynthetic tension, *T* (kN/m) | 33.6 | 41.4 | 26.0 | 115.3 | 9.72 |

3.2.8. GRPS Highway Embankment in China (Liu et al., 2007)

A GRPS embankment was used in this project to support a highway in a northern suburb of Shanghai, China. Pulverized fuel ash was used as an embankment fill. Details of the site conditions and design parameters were reported by Liu et al. [39]: embankment height ($H$) = 5.6 m, unit weight ($\gamma$) = 18.5 kN/m$^3$, oedometric modulus of the subsoil ($E_{oed}$) = 6937 kN/m$^2$, thickness ($D$) = 16 m, pile spacing ($s$) = 3.0 m, pile diameter ($d$) = 1 m, and geosynthetic stiffness ($J$) = 1180 kN/m [40].

This study is similar to the study conducted by Hosseinpour et al. in that the pile diameter determines the cover ratio. The load efficiency may differ in the current method, but the geosynthetic tension should be close to the actual field measurements. Table 13 presents a comparison of the load efficiency and geosynthetic tension. The EBGEO method underestimates the load efficiency, while the results of the CUR 226 and BS 8006 methods align with the measured value. The current method slightly overestimates the geosynthetic tension, which corresponds to the results in Figure 7. The EBGEO method overpredicts the geosynthetic tension compared to the observed value, while the BS 8006 method overestimates it significantly. On the other hand, the CUR 226 method shows good agreement with the measured geosynthetic tension.

**Table 13.** Comparison of the field measurements with design method results for the Liu et al. case study.

| Parameters | Field Measurements | Present Design Method | EBEGO Design Method | BS 8006 Design Method | CUR 226 Design Method |
|---|---|---|---|---|---|
| Load efficiency $E$ (%) | 62.6 | - | 51.1 | 61.4 | 57.9 |
| Geosynthetic tension, $T$ (kN/m) | 19.97 | 26.3 | 53.1 | 281.3 | 21.3 |

## 4. Conclusions

For the development of a geosynthetic-reinforced pile-supported embankment design, a novel method that addresses all interactions between the components of rigid inclusions has been presented. The method employs the finite element method (FEM) to avoid some of the drawbacks of experimental and theoretical methods. In other words, the proposed method overcomes the moot points related to the reality of the soil arch shape, load magnitude carried by the geosynthetic reinforcement, skin friction along the soil–geosynthetic interface, and the role of the soft subsoil and its real behavior.

A full-scale model of the GRPS embankment was validated. Afterward, certain simplifications were made for design purposes, including the adoption of one soft soil layer and the removal of the working platform. The resulting simplified and validated model of the GRPS embankment was then used to collect data, which were analyzed to derive two equations that can be used to determine the load efficiency of the piles and the tension in the geosynthetic. The design method employed six parameters: pile cap width, pile spacing, embankment height, oedometric modulus of the subsoil, geosynthetic stiffness, and embankment fill unit weight. The cover ratio must be governed by adjusting the pile cap area while keeping the pile diameter constant. For the design process, Plaxis 3D and Curve Expert software were utilized in this study.

The present design method is characterized relatively by the simplicity of the solving of the developed equations, and it covers a wide range of the case studies in the literature.

The proposed design method was compared to various design methods, including EBGEO, CUR 226, and BS 8006, as well as to field measurements of load efficiency and geosynthetic tension across eight case studies. The findings indicate that the current method provides reliable results that are consistent with the measured parameters.

## 5. Derivation of an Equation for Load Efficiency (*E*)

The equation below demonstrates how the unit weight of an embankment fill ($\gamma$) impacts the load efficiency ($E$) when considering the constants (a; $E_{oed}$; $J$; $s$).

$$E = \alpha + \alpha_1 * \Delta\gamma \tag{14}$$

In order to account for the influence of the geosynthetic stiffness ($J$) on the load efficiency equation ($E$), it is necessary to determine the correlations between $\alpha$ and $\alpha_1$ (as the dependent variables) and $J$ (as the independent variable). The resulting relationships are presented below:

$$E = (\beta_1 + \beta_2 * J) + \alpha_1 * (\gamma - 19) \tag{15}$$

The given expression below is used to establish the relationship between the embankment height ($H$) and the load efficiency ($E$):

$$E = \left(\frac{\lambda_1 * H^{\lambda_2}}{W_T} + \beta_2 * J\right) + \left(\lambda_3 + \lambda_4 * H + \frac{\lambda_5}{H^2}\right) * (\gamma - 19) \tag{16}$$

$$E = (E_1 + E_2) + E_3 \tag{17}$$

The coefficients are denoted by $\alpha_i$, $\beta_i$, and $\lambda_i$ in Equations (14)–(16). In the present phase of calculation, $E_i$ signifies the components of the load efficiency, while $W_T$ denotes the total weight of the embankment fill and is equal to $(-0.18 + 20.11 * H) * s^{1.97} \approx \gamma_{emb} * H * s^2$.

Owing to some difficulties encountered, a step was carried out by the authors to simplify the derivation process in which the applied load on the pile head $P = \lambda_1 * H^{\lambda_2}$.

In order to determine the influence of the pile diameter (a) on the load efficiency (E), the following equation is provided:

$$E = \left( \frac{\left( \kappa_1 * a^{\kappa_2} * H^{(\kappa_3 + \kappa_4 * a)} \right)}{W_T} + \beta_2 * J \right) + \left( \lambda_3 + \lambda_4 * H + \frac{\lambda_5}{H^2} \right) * (\gamma - 19) \qquad (18)$$

To consider the influence of pile spacing (*s*) on the load efficiency (*E*), the following equation is employed:

$$E = \left( \frac{(F_1 + F_2 * s) * a^{(F_3 * s^{F_4})} * H^{((F_5 + F_6 * s) + \kappa_4 * a)}}{W_T} + \beta_2 * J \right) + \left( \lambda_3 + \lambda_4 * H + \frac{\lambda_5}{H^2} \right) * (\gamma - 19) \qquad (19)$$

Final step involves determining the impact of the elastic modulus ($E_{oed}$):

$$E = \left( \frac{(F_1 + F_2 * s) * a^{(F_3 * s^{F_4})} * H^{((F_5 + F_6 * s) + \kappa_4 * a)}}{W_T} + \beta_2 * J \right) + \left( \lambda_3 + \lambda_4 * H + \frac{\lambda_5}{H^2} \right) * (\gamma - 19)$$

where:

$$
\begin{aligned}
F_1 &= \mu_1 + \mu_2 * E_{oed} \\
F_2 &= \mu_3 + \mu_4 * E_{oed} \\
F_3 &= \mu_5 * E_{oed}{}^{\mu_6} \\
F_4 &= \mu_7 * E_{oed}{}^{\mu_8} \\
F_5 &= \mu_9 + \mu_{10} * E_{oed}
\end{aligned}
\qquad (20)
$$

**Author Contributions:** Conceptualization, R.A.; investigation, A.A.; writing—original draft preparation, R.A.; Numerical modeling, R.A. and A.A.; writing—review and editing, R.A.; supervision, E.K. All authors have read and agreed to the published version of the manuscript.

**Funding:** This research received no external funding.

**Institutional Review Board Statement:** Not applicable.

**Informed Consent Statement:** Not applicable.

**Data Availability Statement:** Data will be made available upon request.

**Conflicts of Interest:** The authors declare no conflict of interest.

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
