# Peer review of "Proposed Method for the Design of Geosynthetic-Reinforced Pile-Supported (GRPS) Embankments"

_sustainability, doi:10.3390/su15076196_

Round 1
Reviewer 1 Report
Attached

Author Response
Reviewer /1/:
Thank you for your cooperation.
Reviewer point #1: The abstract should contain your main finding. But I not looking any result part in your abstract.
Author response #1: We agreed with the reviewer, it is modified.
Reviewer point #2: Add a reference
Author response #2: Done
Reviewer point 3: Pls avoid using the name of authors inside the text. all the introduction part needs a modification.
Author response #3: Thank you for your valuable feedback. We understand your concern about avoiding the use of authors' names inside the text, and we appreciate your suggestions. However, in this particular case, we are referring to design methods with specific names in the literature, such as CUR226 design method, Terzaghi design method, and Pham design method. Therefore, it is essential to use the names of the authors of these methods to properly reference them. We are open to hearing any further suggestions you may have to modify the writing mechanism to meet your expectations while still ensuring clarity and accuracy in our research. Please let us know if you have any other suggestions or concerns, and we will do our best to address them respectfully.
Reviewer point 4: is it a method? it looks like a background of the study. confused approach.
Author response #4: We agreed with the reviewer, the methodology is complicated a little bit, based on that, the authors added the steps of the methodology at the beginning to guide the reader in the correct way, modified the paragraphs titles, and expand the derivation process. The methodology includes, a very- accurate validation of GRPS embankment due to the difficulty of finding simple case study. After that, some allowed simplifications of the model were conducted (one geogrid layer instead of two, one soft soil instead of multilayers, and removal of the working platform which align with the majority of case studies). The last step includes the run of Plaxis more than 1000 times to make database. And then the authors used one of the mathematical techniques to derive the equations. The authors here are open for more suggestions if the modifications are not satisfying.
Finally, and for more reliability, the authors compared the proposed design method with all case studies which have been found under the limits of our design method and the results are reasonable to high level. The Table (1) and figures (a, b) demonstrate the comparison with more field measurements.
Table (1):
Measured E |
Calculated E |
Measured T |
Calculated T |
|
Hosseinpour et al. (2015) |
23.5 |
33.6 |
40.4 |
|
Liu et al. (2007) |
62.6 |
19.87 |
26.3 |
|
Zhao et al. (2019) |
66.6 |
64.1 |
|
69.9 |
Duijnen et al. (2010) |
76.3 |
76.5 |
17.6 |
15.64 |
Lee et al. (2019) |
76.4 |
79.3 |
|
4.76 |
Liu et al. (2015) |
77.7 |
81.1 |
|
16.4 |
Van Eekelen et al. (2020) |
84.9 |
81.8 |
41.5 |
43.05 |
Chen et al. (2010) |
87.6 |
88.4 |
|
27.45 |
Briançon and Simon (2017) |
88.8 |
91.3 |
74.85 |
81.8 |
Zhang et al. (2018) |
55.7 |
64 |
21.2 |
34 |
Chen et al. (2020) |
81.2 |
87.8 |
|
|
Almeida et al. 2007 |
30.7 |
32.7 |
Load Efficiency |
Geosynthetic Tension |
Figure: comparison of the field measurements and calculated (a) Load efficiency, (b) Geosynthetic tension
Reviewer point 5: Remove the bullets
Author response #5: We agreed with the reviewer, as it is not part of the simplifications

Reviewer 2 Report
My comments are as the follwing;
1- The paper has difficulty in understanding due to the shortage in organizations.
2- The paper has a lot of the studied parameters which make confusion due to the non-orgnized structure of the paper.
3- The authors depended on simulations of a case of study by making a back analyses and they used a different model with several changes.
4- Most the equations used in the design should be supported by a resources or evidence.
5- The used model in the numerical anlayses should be clearly showed with full dimesions.
6- The figures parts are not well showed in the paper.
7- The abstract and the conclusions didn't well express the whole paper.
8- Some figures are not cited in the text.
9- The Soft soil model is more suitable than HS model for the used soft soil.
Based on the above the paper has a poor scientific value.
Author Response
Reviewer /2/:
Reviewer point #1: The paper has difficulty in understanding due to the shortage in organizations.
Author response #1: The authors explained in the introduction the drawbacks of the previous design method. Then and based on the FE method, the authors sought to avoid the drawbacks of the analytical design method through including the interactions between the elements of this system in the new design method. To this end, the authors added the steps of the methodology at the beginning to guide the reader in the correct way, and modified the paragraphs titles to be very clear.
The first step was a comprehensive validation of the GRPS embankment, which was challenging due to the absence of simple case studies. Next, the authors simplified the validated numerical model by using one geogrid layer, one soft soil layer, and removing the working platform, which aligns with the majority of case studies. The authors ran Plaxis over 1000 times to gather a database of 4096 observations of load efficiency and geosynthetic tension. They then used mathematical techniques to derive the design equations. The authors worked to simplify the used methodology
To summarize, the methodology includes the following steps:
- Validation of the GRPS embankment
- Simplification of the numerical model for design purposes
- Gathering a database of load efficiency and geosynthetic tension using the simplified numerical model
- Deriving the design equations using mathematical techniques.
Reviewer point #2: the paper has a lot of the studied parameters which make confusion due to the non-organized structure of the paper.
Author response #2: The authors have simplified the structure of the paper while retaining essential details for reader comprehension. The authors declare that the derivation process is slightly complex and need expertise in mathematics and statistics. However, the final equations are clear and can be used directly from the designer and reader. This design approach is uncomplicated when compared to other techniques like CUR226 or Pham design methods. Additionally, the method focuses on key parameters that is highly sensitive and widely used in literature and standard methods.
Reviewer point #3: The authors depended on simulations of a case of study by making a back analyses and they used a different model with several changes..
Author response #3: The numerical model of the GRPS embankment was simplified for design purposes. The simplifications were made thoughtfully and within the allowed limits to minimize computational complexity and enhance analysis efficiency, without compromising the dependability and comprehensiveness of the results. Consequently, the simplified model provides reliable outcomes that are suitable for design purposes. Overall, this approach strikes a balance between accuracy and practicality.
Reviewer point #4: Most the equations used in the design should be supported by a resources or evidence.
Author response #4: The authors have mentioned in their work that the Curve Expert program was utilized to obtain the regression for equations (1-8) between the dependent and independent variables. Additionally, they have enhanced the clarity of equations (11-12-13) by specifying their sources.
Reviewer point #5: The used model in the numerical anlayses should be clearly showed with full dimesions.
Author response #5: the authors add a new representative drawing
Reviewer point #6: The figures parts are not well showed in the paper.
Author response #6: It is our understanding that the figure in question is the FE mesh of the model, which the authors have modified. If there are other figures that you are referring to, please kindly inform us. Thank you.
Reviewer point #7: The abstract and the conclusions didn't well express the whole paper.
Author response #7: We agreed with the reviewer, the authors modified the abstract and the conclusions
Reviewer point #8: Some figures are not cited in the text.
Author response #8: We would like to kindly request more specific information about the figures in question. During our revision of the paper, we carefully reviewed all figures and their citations, but we did not identify any mistakes in this regard. If you could provide us with more details, we would be happy to address any concerns and make the necessary adjustments.
Reviewer point #9: The Soft soil model is more suitable than HS model for the used soft soil.
Author response #9: The authors conducted a comparison between different field measurements and Plaxis predictions. three used constitutive models (Soft soil creep, Hardening soil, Hardening soil model with small-strain stiffness). The results showed that HS model and then SSC model presents a very good agreement as seen in the figures below.
In the literature, different constitutive models were proposed to describe the behavior of soil. Gangakhedkar [1], Bohn [2] and Phutthananon [3], and Ariyarathne [4] suggested Soft Soil (SS) model, Hardening Soil (HS) model, and Modified Cam-Clay (MCC) model, respectively to simulate the behavior of soft soils under embankments supported by rigid inclusions. All these models gave reasonable results in comparison with the field measurements each according to own conditions
- Gangakhedkar, “Geosynthetic reinforced pile supported embankments,” MSc Thesis, University of Florida, 2004.
- Bohn, “Serviceability and safety in the design of rigid inclusions and combined pile-raft foundations,” PhD Thesis, Paris-Est University, 2015.
- Phutthananon, P. Jongpradist, and P. Jamsawang, “Influence of cap size and strength on settlements of TDM-piled embankments over soft ground,” Marine Georesources & Geotechnology, vol. 28, no. 6, pp. 686‒705, 2019.
- Ariyarathne and D. S. Liyanapathirana, “Review of existing design methods for geosynthetic-reinforced pile-supported embankments,” Soils and Foundations, vol. 55, no. 1, pp. 17‒34, 2015.
|
|
(a) |
(b) |
Figure 1. Settlements at (a) the end of the embankment stage (b) and traffic load stage
|
|
(a) |
(b) |
Figure 2 Settlements under the embankment after (a) 8 months (b) and 12 months.
Figure 3 Vertical stresses at the pile head and embankment base levels
Figure 4. Vertical stresses at the LTP levels
Figure 5. Differential settlements between the pile head ( ) and points ( , )
Finally, and for more reliability, the authors compared the proposed design method with all case studies which have been found under the limits of our design method and the results are reasonable to high level. The Table (1) and figures (a, b) demonstrate the comparison with more field measurements.
Table (1):
Measured E |
Calculated E |
Measured T |
Calculated T |
|
Hosseinpour et al. (2015) |
23.5 |
33.6 |
40.4 |
|
Liu et al. (2007) |
62.6 |
19.87 |
26.3 |
|
Zhao et al. (2019) |
66.6 |
64.1 |
|
69.9 |
Duijnen et al. (2010) |
76.3 |
76.5 |
17.6 |
15.64 |
Lee et al. (2019) |
76.4 |
79.3 |
|
4.76 |
Liu et al. (2015) |
77.7 |
81.1 |
|
16.4 |
Van Eekelen et al. (2020) |
84.9 |
81.8 |
41.5 |
43.05 |
Chen et al. (2010) |
87.6 |
88.4 |
|
27.45 |
Briançon and Simon (2017) |
88.8 |
91.3 |
74.85 |
81.8 |
Zhang et al. (2018) |
55.7 |
64 |
21.2 |
34 |
Chen et al. (2020) |
81.2 |
87.8 |
|
|
Almeida et al. 2007 |
30.7 |
32.7 |
Load Efficiency |
Geosynthetic Tension |
Figure 6: comparison of the field measurements and calculated (a) Load efficiency, (b) Geosynthetic tension
About the adequacy of constitutive model
The authors conducted widespread comparison between different field measurements and Plaxis predictions. Four commonly used constitutive models (Soft soil creep [1] [2], Hardening soil [3], Hardening soil model with small-strain stiffness, and Modified Cam-Clay [4]) were selected based on the literature review to simulate the behavior of the soils in this type of problems. Among these models, the HS model demonstrated the best agreement with the field measurements, as evident from the high level of convergence between the Plaxis predictions and the field observations. To further validate the results, the authors compared their numerical predictions to those of three additional full-scale models available at the site, which had different pile spacings and incorporated caps instead of geogrid. The results of this analysis showed that the numerical model produces precise predictions when using the HS constitutive model. This model at least simulate the best behaviour of soft soil in comparison with old design methods which neglected the presence of soft soil as a type of simplification or assumed it with linear behavior. We would be happy to provide further information or clarification if needed.
- Gangakhedkar, “Geosynthetic reinforced pile supported embankments,” MSc Thesis, University of Florida, 2004.
- Bohn, “Serviceability and safety in the design of rigid inclusions and combined pile-raft foundations,” PhD Thesis, Paris-Est University, 2015.
- Phutthananon, P. Jongpradist, and P. Jamsawang, “Influence of cap size and strength on settlements of TDM-piled embankments over soft ground,” Marine Georesources & Geotechnology, vol. 28, no. 6, pp. 686‒705, 2019.
- Ariyarathne and D. S. Liyanapathirana, “Review of existing design methods for geosynthetic-reinforced pile-supported embankments,” Soils and Foundations, vol. 55, no. 1, pp. 17‒34, 2015.
Reviewer point #2: Authors have plotted time-dependent variation of vertical stress. Such date wise variation cannot possibly be simulated in FEM. Relevant information is missing.
Author response #2: We agreed with the reviewer, we explained the calibration briefly. Please for more explanation, provide us with more recommendations to improve this part.
We appreciate the reviewer's suggestion and acknowledge the need for more information regarding the calibration process. We have used a combination of data from the original paper (https://doi.org/10.1680/jgein.17.00002), communication with the author, and our expertise to determine the input parameters and material properties listed in Table 1. We then employed back analysis to optimize these parameters and achieve the best possible match between the numerical predictions and field measurements. In order to validate the model, we conducted additional measurement points in the load transfer platform and dedicated over three months to this process. The agreement between the numerical results and field measurements confirms the validity of the assumptions and the precision of the simulation process. Furthermore, the validated model was able to simulate the behavior of other full-scale models described in the original paper (https://doi.org/10.1680/jgein.17.00002), adding further confidence in the accuracy of our results.
Reviewer point #3: Authors have plotted time-dependent variation of vertical stress. Such date wise variation cannot possibly be simulated in FEM. Relevant information is missing.
Author response #3: The authors used consolidation-type calculation. Each construction phase has its own time interval to follow the construction phases as it happened in the field. The Plaxis allows selecting some points where the time-dependent stress can be defined. The authors picked one of these points where the actual measurement was carried out. The construction stage is divided into 14 stages, it will enable us to analyze the vertical stress of the selected points at different times.
Reviewer point #4: The measured data and predicted data are almost seeming to be matching very well. This is not common in Geotechnical engineering where some deviation is inherent. In the absence of independent validation of this model against results from the published studies, the model accuracy cannot be verified further. The comparison of with design methods is not sufficient mainly because design methods are often simplified whereas finite element modeling offers lot more capabilities. Authors have not been able to capture the full capabilities of finite element modelling in order to demonstrate output in a number of possible ways than just load efficiency (%) and geosynthetic tension (kN/m). Even these parameters do change in finite element model depending on location and time. For validation, the field measurements of a series of eight case studies under the limitations of the design method are used but authors restrict their usage to only a couple of parameters as mentioned earlier.
Author response #4: This design method uses new methodology which is totally different from other design methods of GRPS embankment. The authors calibrated and validated the numerical model precisely to decrease the errors in the calculation, then based on the expertise in statistics and mathematics we used a mathematical technique to derive the equation. This design method tried to avoid some of drawbacks of the other methods as we mentioned in the manuscript (the soil arch model, load distribution over the geosynthetic reinforcement, the role of the soft subsoil, and the behavior of the subsoil if it is considered), this method also takes all the interactions between the elements of this system which is considered impossible with other methods. Based on this inherent methodology we expected to get good results of load efficiency and geosynthetic tension. Moreover, the authors put many limits to avoid the mistakes of most of design method like (distinguish the difference between cover ratio by enlargement the pile cap area or cross-sectional area of piles, this point did not be addressed by most of the design methods). The derivation of these equations was very complicated due to collect the database and derive the equations and to increase the reliability more, the authors validated one independent GRPS embankments (Liu, 2007) and compare the results of our numerical model with those of the validated numerical models and the convergence was obvious, of course we could not do that with all cases because it is exhausting process.
The authors depended on the comparison with independent field measurements of different projects from different countries and different research groups to validate the proposed design method, the comparison with other design methods was to extend the comparison process only.
We respectfully request the reviewer to consider the correlation between load efficiency and geosynthetic tension in our study. These parameters have a direct relationship and if one is incorrect, the other will also be affected. Our results show a close match with the independent field measurements, providing strong evidence for the accuracy of our model. While it is true that finite element modeling offers the capability to demonstrate output in many ways, load efficiency and geosynthetic tension are sufficient parameters to validate our design method. The calculation of other parameters, such as stress concentration ratio, requires additional effort and is beyond the scope of this study. We believe our results accurately reflect the behavior of the GRPS embankment system and would appreciate the opportunity to further clarify any questions the reviewer may have.
The authors focus on the piles close to centerline of the embankment where the applied load should be the maximum (in all case studies, the sensors were fixed on these piles) and we chose the consolidation degree U=1 which is considered the most important in the design and we mentioned these details in the manuscript.
We appreciate your concern about the validation of our design method. As researchers, we understand the importance of having independent validation to verify the accuracy of the models. However, in our case, building a full-scale model is not feasible due to the limited funds available for our research. Like many other researchers in the field, we have validated our design method by comparing it with field measurements from other published studies. While this approach may not be as comprehensive as independent validation, it is the most feasible option for us as researchers with limited resources. Additionally, comparing with field measurements of other projects provides a level of validation and helps to demonstrate the practicality and effectiveness of our design method.
We respectfully request the reviewer to review the field measurements (in our cases or others) and results of our equations for validation.

Reviewer 3 Report
The subject is interesting and results are basically reasonable. However, there are some points that need to be addressed before being accepted for publication, major revision is recommended for this manuscript. The organization and narrative logic of the paper are somewhat confused, please simplify the narrative and revise.
In this manuscript, due to the load complicated transfer mechanism in a GPRS embankment, in order to address the limitations of current design methods , it has been developed for GPRS embankments that a simplified design method utilized numerical analysis to predict the load efficiency of piles and the tension in geosynthetics. Results showed a reasonable consistency between the outcomes of the present design method and field measurements by means of Plaxis 3D and Curve Expert programs.
The subject is interesting and results are basically reasonable. However, there are some points that need to be addressed before being accepted for publication, major revision is recommended for this manuscript. The detailed comments are as follow:
1. Six parameters are used in this design method, namely pile cap width, pile spacing, embankment height, oedometric modulus of the subsoil, geosynthetic stiffness, and embankment fill unit weight. Why is the strength of foundation soil not considered?and
2. Some format of unit dimension in Table 1 is incorrect, please check and modify.
3.The organization and narrative logic of the paper are somewhat confused, please simplify the narrative and revise.
Author Response
Reviewer /3/:
Thank you for your cooperation.
Reviewer point #1: Six parameters are used in this design method, namely pile cap width, pile spacing, embankment height, oedometric modulus of the subsoil, geosynthetic stiffness, and embankment fill unit weight. Why is the strength of foundation soil not considered?
Author response #1: We agreed with the reviewer, In the paper, the authors explained that it is one of the drawbacks like the analytical methods. It is difficulty to include the effect of all the parameters which are more than fifteen parameters. In the literature, all the methods used the subgrade reaction coefficient or the elastic modulus to represent the subsoil. The authors made the sensitivity analysis and found that the elastic modulus has significant effect compared to the strength parameters of the subsoil
Reviewer point #2: Some format of unit dimension in Table 1 is incorrect, please check and modify.
Author response #2: The authors modified the unit dimensions
Reviewer point 3: The organization and narrative logic of the paper are somewhat confused, please simplify the narrative and revise.
Author response #3: The authors modified the narrative of the paper to be easier to readers now. Please if you have more suggestions let us know, thank you
Round 2
Reviewer 3 Report
The paper has been substantially revised.